# Corticosteroids for the Treatment of Internal Temporomandibular Joint Disorders: A Systematic Review and Network Meta-Analysis

**DOI:** 10.3390/jcm13154557

**Published:** 2024-08-04

**Authors:** Daniela Torres, Carlos Zaror, Verónica Iturriaga, Aurelio Tobias, Romina Brignardello-Petersen

**Affiliations:** 1Magíster en Odontología, Facultad de Odontología, Universidad de La Frontera, Temuco 4781176, Chile; danielatorresbascur@gmail.com; 2Temporomandibular Disorder and Orofacial Pain Program, Sleep & Pain Research Group, Faculty of Dentistry, Universidad de La Frontera, Temuco 4781176, Chile; veronica.iturriaga@ufrontera.cl; 3Department of Pediatric Dentistry and Orthodontics, Faculty of Dentistry, Universidad de La Frontera, Temuco 4781176, Chile; 4Department of Health Research Methods, Evidence, and Impact, McMaster University, Hamilton, ON L8S 4L8, Canada; brignarr@mcmaster.ca; 5Department of Integral Adult Care Dentistry, Faculty of Dentistry, Universidad de La Frontera, Temuco 4781176, Chile; 6Institute of Metabolism and Systems Research, University of Birmingham, Birmingham B15 2TT, UK; aurelio.tobias@gmail.com

**Keywords:** temporomandibular joint disorders, corticosteroids, systematic review

## Abstract

**Background:** We evaluated the comparative effectiveness of all intra-articular injection corticosteroids for treating internal temporomandibular joint (TMJ) disorders. **Methods**: We searched MEDLINE, CENTRAL, EMBASE, SCOPUS, and LILACS through December 2023. We included randomized clinical trials (RCTs) enrolling patients with symptomatic internal disorders of the TMJ comparing any type of intra-articular corticosteroid therapy against another or to another minimally invasive therapy. The outcomes of interest were pain, range of mandibular motion (RoM), quality of life (QoL) and adverse effects at 1, 3, 6, and 12 months. We assessed the risk of bias using the Cochrane Collaboration’s tool. We conducted a frequentist network meta-analysis and assessed the certainty of the evidence (CoE) using GRADE. **Results**: We included 20 RCTs enrolling 810 participants, which assessed five corticosteroids alone or combined with arthrocentesis or hyaluronic acid. Based on moderate CoE, betamethasone is among the most effective corticosteroids for reducing pain at one (mean difference compared to arthrocentesis [MD], −3.80; 95% confidence interval [CI], −4.55 to −3.05) and three months (MD, −2.74; 95%CI, −3.42 to −2.06), and arthrocentesis plus dexamethasone at six months (MD, −0.80; 95%CI, −1.57 to −0.03). There was no convincing evidence that any intervention was better than arthrocentesis for improving the RoM and QoL at any follow-up time. Methylprednisolone may be more harmful than arthrocentesis for adverse effects. **Discussion**: Betamethasone and arthrocentesis plus dexamethasone are the most effective in managing pain in the short and medium term compared to arthrocentesis (moderate CoE). Decisions about their use should consider other factors, such as costs, feasibility, and acceptability. Future research should consider QoL as an outcome and assess participants at longer follow-up periods.

## 1. Introduction

Temporomandibular disorders (TMDs) involve a broad group of pathologies that affect the temporomandibular joint (TMJ) and its adjoining structures [1]. TMDs are a significant public health problem affecting approximately 5% to 12% of the overall population [2] and considered the most common cause of chronic pain of nondental origin in the orofacial area [1]. The prevalence of TMDs in the world population is estimated at 34% and is higher in South America (47%) compared to Asia (33%) and Europe (29%). The age group of 18 to 60 years is the most exposed to TMD, with a higher ratio of females than males [3].

TMJ disorders include extra- and intracapsular disorders. Intracapsular or internal TMJ disorders correspond to approximately 80% of symptomatic TMDs, the most frequent being condyle–disc disorders and osteoarthritis [4].

Treatment of internal TMJ disorders may range from conservative (counseling about the etiologic causes to providing support regarding postural and behavioral habits, physiotherapy, occlusal splints, and anti-inflammatory drugs) and minimally invasive approaches (arthrocentesis or intra-articular injections of corticosteroids and hyaluronic acid, among others) to more invasive therapies, such as discectomy, high condylectomy, and arthroplasty. All of these interventions seek to relieve pain, restore jaw function, and prevent disease progression [5,6].

Corticosteroids are one of the most widely used minimally invasive therapies to manage internal TMJ disorders. Due to their anti-inflammatory effects, they have shown positive results in relieving pain in different joints [7]. Corticosteroids inhibit the production and secretion of proinflammatory cytokines and prevent the accumulation of macrophages and neutrophils in inflammatory foci [8,9]. The most commonly used corticosteroids for the intra-articular management of different joints are hydrocortisone, methylprednisolone, dexamethasone, betamethasone, prednisolone, and triamcinolone [10].

Although some systematic reviews compare the different minimally invasive therapies for internal disorders of the TMJ [11,12,13], to our knowledge, there are no systematic reviews that compare the effectiveness of the different intra-articular corticosteroids with one another, thus making it difficult to determine which is the most beneficial for these internal TMJ disorders. Therefore, we conducted a systematic review and network meta-analysis (NMA) to assess the comparative effects of intra-articular corticosteroids on pain management, range of mandibular motion (RoM), quality of life (QoL), and adverse effects in patients with internal TMJ disorders.

## 2. Materials and Methods

### 2.1. Protocol and Registration

We report this systematic review and NMA following the preferred reporting items for systematic reviews and meta-analyses (PRISMA) checklist for NMA [14] (see Appendix A). We registered the protocol in PROSPERO (CRD42019129014) and published it in an open-access journal [15].

### 2.2. Eligibility Criteria

We included randomized controlled clinical trials (RCTs) enrolling patients with any symptomatic internal TMJ disorders, comparing any dose and regimen of intra-articular corticosteroid injections with one another or to another minimally invasive therapy, such as intra-articular injection of hyaluronic acid, platelet-rich plasma, saline, or Ringer’s lactate solution. Outcomes of interest were pain measured through a visual analogue scale from 0 to 10, RoM evaluated by measuring the mouth opening in millimeters, QoL measured using a validated scale, and adverse effects. Time points of interest were 1, 3, 6, and 12 months. We included studies published in English, Spanish, French, Portuguese, and German languages. We excluded intra-articular injections used for temporomandibular muscle disorders like botulinum toxin and anesthetic, among others.

### 2.3. Information Sources and Search

We conducted electronic searches in MEDLINE, CENTRAL, EMBASE, SCOPUS, and LILACS from their inception to 31 December 2023. The details of the search strategy used in MEDLINE (PubMed) are listed in an online Appendix A. In addition, we searched OpenGray, ClinicalTrials.gov, and the WHO International Clinical Trials Registry. We also examined the reference list of included studies and other published systematic reviews.

### 2.4. Study Selection

Two reviewers (DT and VI) independently reviewed the titles, abstracts, and full text using the software COVIDENCE (www.covidence.org; accessed 23 July 2024) to identify relevant studies. A third reviewer (CZ) resolved discrepancies.

### 2.5. Data Collection Process and Data Items

For each eligible trial, a pair of reviewers (DT and VI), following training and calibration exercises, extracted data independently using a standardized, pilot-tested data extraction form. Extracted information included study design/setting, study population, participant demographics, baseline characteristics, details of the intervention and comparator, outcome data, and follow-up times. A third reviewer (CZ) resolved disagreements between the review authors. We contacted the authors for clarification or missing information. We used WebPlotDigitizer 4.2 for Mac (https://automeris.io/WebPlotDigitizer; accessed 23 July 2024) to extract data presented graphically.

### 2.6. Risk of Bias within Individual Studies

Two reviewers (DT and VI) independently assessed the risk of bias in the included studies using the Cochrane Risk of Bias Tool (RoB 2.0) [16]. RoB 2.0 evaluates the following domains: bias arising from the randomization process, bias due to deviations from intended interventions, bias due to missing outcome data, bias in measurement of the outcome, and bias in selection of the reported results. For each study, we judged each domain at low risk of bias, some concerns, or high risk of bias. We also judged the study-level risk of bias at low risk of bias, some concerns, or high risk of bias using the guidance provided by the tool authors, where studies with a high risk of bias in at least one domain or studies with some concerns for multiple domains are considered at high risk of bias at the study level. Reviewers resolved discrepancies through adjudication by a third party (CZ).

### 2.7. Data Synthesis

We conducted frequentist fixed-effects NMA using multivariate meta-analysis using Stata 18 (Stata Corp LP, College Station, TX, USA) because we assumed that all studies share a single common effect, as we had few studies for each comparison. We created network plots to illustrate the network geometry. The node’s size represents the number of patients who received the treatment, and the thickness of the edges represents the number of trials directly comparing the two interventions. We grouped the treatments into common nodes based on the type of corticosteroid and not on the dose. We created separate nodes for those interventions that combined corticosteroids with other minimally invasive therapies. The reference for all NMAs was arthrocentesis, which is the extraction of synovial fluid from the joint cavity with or without lavage with saline or Ringer’s lactate solution. For each comparison, we assessed incoherence by comparing direct and indirect estimates, which we obtained using the node-splitting technique [17].

If data were insufficient for an NMA, we conducted a frequentist pairwise meta-analysis (PMA) comparing specific interventions. We carried out the PMA using the inverse-variance method and a fixed-effect model.

For continuous outcomes (pain, RoM, and QoL), we estimated the effects of interventions using the mean difference with their corresponding 95% confidence interval (CI). For binary outcomes (adverse effects), we used the relative risk (RR) and its corresponding 95%CI. Then, we calculated the risk difference based on using the RR and baseline risks obtained from the median of control arms of the included studies.

When studies reported the same outcome using a scale with a different range, before conducting analyses, we converted data to the scale range most commonly reported [18]. The original authors of the study were contacted to obtain the missing data and details of any outcomes that may have been measured but not reported. We did not use any other statistical methods or perform any further imputation to account for missing data.

We assessed the clinical heterogeneity of the included studies by examining the similarity between the types of participants, interventions, and outcomes. In pairwise meta-analyses, we estimated the statistical heterogeneity for each comparison using the I² statistic, where values over 50% indicate considerable heterogeneity [16]. In the network meta-analysis, we assumed a common estimate for the heterogeneity variance across all comparisons.

We planned to use funnel plots to assess publication bias when the number of studies pooled was >10.

### 2.8. Additional Analyses

We planned to conduct a subgroup analysis according to the type of symptomatic internal TMJ disorders, QoL questionnaires (health-related quality of life instruments versus oral-health-related quality of life instruments), and the duration of follow-up as a possible source of heterogeneity between studies.

We conducted a sensitivity analysis for each outcome to assess the robustness of the findings by excluding studies with an overall high risk of bias according to the RoB2 assessment.

### 2.9. Certainty of Evidence and Reporting Conclusions

We used the Grading of Recommendations, Assessment, Development, and Evaluations (GRADE) working group’s approach for rating the certainty of the NMA evidence for all of the comparisons and outcomes as high, moderate, low, or very low [19,20]. The GRADE approach assesses the certainty of evidence from NMA by considering risk of bias, inconsistency, indirectness, publication bias, imprecision, intransitivity, and incoherence. The assessment follows a stepwise approach in which both direct and indirect evidence, and their contribution to the NMA estimate, determine the certainty of evidence [21].

We used a minimally contextualized approach, with a null effect as the threshold to assess imprecision [22].

To draw conclusions for each outcome, we classified interventions from the most to the least effective by considering the estimates of effect and the certainty of the evidence [22] and presented the results graphically to facilitate their interpretation [23].

## 3. Results

### 3.1. Study Selection

After identifying 1243 titles, eliminating duplicates, and reviewing titles and abstracts, we evaluated 34 full-text articles. We excluded five articles because they were not RCTs [24,25,26,27,28], four because they did not compare corticosteroids with arthrocentesis or minimally invasive therapy [29,30,31,32], and one because it did not include appropriate outcomes [33]. Appendix A provides the reasons for excluding full-texts reviewed at this stage. Finally, we included 24 articles corresponding to 20 studies because 3 studies were reported in more than 1 article (See Figure 1).

### 3.2. Study Characteristics

Table 1 presents the main characteristics of the included studies. Studies were published between 1985 and 2022, and half were conducted in India, Iraq, and Turkey. All trials had a parallel-group design and enrolled 10 to 102 participants, with a mean age of 39.24 (SD = 8.93); most participants were female (77.72%). The internal TMJ disorders included disc displacements with reduction (six trials), disc displacements without reduction (four trials), arthralgia (two trials), and osteoarthritis (eight trials). Five trials did not specify the internal TMJ disorders included. The corticosteroids evaluated were betamethasone, dexamethasone, methylprednisolone, hydrocortisone, and triamcinolone alone or combined with arthrocentesis or hyaluronic acid. Trials reported funding from academic or government institutions [23,24,25,26,27,28].

### 3.3. Risk of Bias within Studies

Appendix A presents the risk of bias assessment of the included studies for each outcome. Only three studies had a low risk of bias. The domains with more concerns were bias arising from the randomization process because studies did not provide information on allocation concealment and bias in the measurement of the outcome because the outcome assessors were not blinded.

### 3.4. Effects of the Interventions

We conducted an NMA for the outcomes pain, RoM, and adverse effects and a PMA for the outcome QoL. Figure 2 summarizes the effects of interventions for pain and the RoM. Appendix A shows the absolute effect estimates and the certainty of the evidence for all comparisons and outcomes.

#### 3.4.1. Network Geometry

Figure 3 depicts the network plot for pain and RoM at 1, 3, and 6 months. The number of interventions ranged from 7 to 13 across outcomes. Although in most of the networks there was direct evidence for some comparisons that did not include the reference, many estimates came only from indirect evidence or were estimated from a single trial.

#### 3.4.2. Pain

##### Pain at One Month

We included ten trials enrolling 395 participants in the NMA. Based on our classification, betamethasone is among the most effective (measured with a VAS scale from 0 to 10, MD −3.8; 95%CI −4.55 to −3.05 compared with arthrocentesis; moderate-certainty evidence), and triamcinolone and arthrocentesis plus hyaluronic acid and betamethasone may be among the most effective interventions in reducing pain at one month (MD −5.52; 95%CI −6.56 to −4.48 and MD −2.29; 95%CI −4.07 to −0.52, compared with arthrocentesis, respectively; low certainty of evidence). Arthrocentesis plus dexamethasone, arthrocentesis plus methylprednisolone, and arthrocentesis plus triamcinolone may be among the least effective interventions in reducing pain at one month compared with arthrocentesis. For the rest of the interventions, the effect is uncertain because the certainty of evidence is very low.

##### Pain at Three Months

We included nine trials enrolling 307 participants. Betamethasone is among the most effective (MD −2.74; 95%CI −3.42 to −2.06 compared with arthrocentesis; moderate-certainty evidence), and arthrocentesis plus hyaluronic acid and betamethasone may be among the most effective interventions in reducing pain at three months (MD −2.51; 95%CI −4.51 to −0.51 compared with arthrocentesis; low certainty of evidence). A low certainty of evidence showed that triamcinolone was among the interventions with intermediate effectiveness in reducing the pain (MD −0.88; 95%CI −1.72 to −0.04, compared with arthrocentesis). Arthrocentesis plus methylprednisolone and arthrocentesis plus triamcinolone may be among the least effective interventions in reducing pain at one month compared with arthrocentesis. For the rest of the interventions, the effect is uncertain because the certainty of evidence is very low.

##### Pain at Six Months

We included five trials enrolling 162 participants. Arthrocentesis plus dexamethasone is among the most effective interventions in reducing pain at six months compared with arthrocentesis (MD −0.80; 95%CI −1.57 to −0.03; moderate-certainty evidence). Arthrocentesis plus methylprednisolone may be among the least effective interventions in reducing pain at one month compared with arthrocentesis. For the rest of the interventions, the effect is uncertain because the certainty of evidence is very low.

##### Pain at Twelve Months

One study with 24 participants compared arthrocentesis plus methylprednisolone versus arthrocentesis. A pairwise comparison showed there is uncertainty whether methylprednisolone reduces the pain at 12 months when compared with arthrocentesis (MD 0.65; 95%CI −2.41 to 3.72; very low-certainty evidence).

#### 3.4.3. Range of Mandibular Motion

##### RoM at One Month

Eight studies with 310 participants reported sufficient data to assess RoM in this period. However, there was no convincing evidence that arthrocentesis plus betamethasone, arthrocentesis plus dexamethasone, arthrocentesis plus methylprednisolone, arthrocentesis plus triamcinolone, or hyaluronic acid plus triamcinolone differed from arthrocentesis. For arthrocentesis plus hyaluronic acid and betamethasone, the effect is uncertain because the certainty of evidence is very low.

##### RoM at Three Months

Eight studies with 270 participants reported sufficient data to assess the RoM at 3 months. However, there was no convincing evidence that arthrocentesis plus betamethasone, arthrocentesis plus methylprednisolone, or hyaluronic acid plus triamcinolone differed from arthrocentesis. For the rest of the interventions, the effect is uncertain because the certainty of evidence is very low.

##### RoM at Six Months

Five studies with 162 participants reported sufficient data to assess the RoM at 6 months. However, there was no convincing evidence that arthrocentesis plus dexamethasone or arthrocentesis plus methylprednisolone differed from arthrocentesis. For the rest of the interventions, the effect is uncertain because the certainty of evidence is very low.

##### RoM at Twelve Months

One study with 24 participants showed there is uncertainty about whether methylprednisolone plus arthrocentesis increases the range of motion at 12 months when compared with arthrocentesis (MD 0.67; 95%CI −4.88 to 6.22; very low-certainty evidence).

#### 3.4.4. Quality of Life

##### QoL at One Month

One trial with 54 participants, comparing methylprednisolone with arthrocentesis, assessed QoL at one month using the Mandibular Function Impairment Questionnaire (MFIQ, range 0 to 68, higher scores represent worse QoL). Methylprednisolone may not improve the QoL compared with arthrocentesis (MD −0.30; 95%CI −1.26 to 0.66; low-certainty evidence).

##### QoL at Six Months

One trial, including 28 participants, assessed QoL at six months using the MFIQ. A pairwise comparison showed arthrocentesis plus dexamethasone make no difference in improving the QoL compared with arthrocentesis (MD −7.2; 95%CI −20.1 to 5.7; low-certainty evidence).

#### 3.4.5. Adverse Effects

Five studies with 240 participants suggest that methylprednisolone may be more harmful than arthrocentesis (RD 180 more per 1000; 95%CI 16 more to 322 more; low-certainty evidence). The adverse events reported were an increase in pain, paresthesia of the eyelid, numbness, rash, difficulties opening the jaw, TMJ sounds, and headache.

For the rest of the interventions, the effect is uncertain because the certainty of evidence is very low.

### 3.5. Results of Additional Analyses

Because we had few studies, we did not conduct subgroup analysis according to the type of symptomatic internal TMJ disorders or QoL questionnaires.

Sensitivity analysis, excluding high-risk bias studies, showed a similar effect to the main analysis.

Finally, as we had no more than ten studies to pool in any analysis, we did not construct funnel plots to assess publication bias.

## 4. Discussion

This NMA included 20 studies that enrolled 810 participants. We found moderate-certainty evidence that betamethasone was among the most effective interventions for reducing pain at 1 and 3 months compared with arthrocentesis. Based on moderate-certainty evidence, only arthrocentesis plus dexamethasone was among the most effective interventions for reducing pain at six months compared with arthrocentesis.

There was no convincing evidence that any corticosteroids alone or combined with arthrocentesis or hyaluronic acid differed from arthrocentesis for improving the range of motion at any follow-up time.

There is uncertainty as to whether the corticosteroids improve the QoL at one or six months, and methylprednisolone was the only corticosteroid shown to have more adverse effects than arthrocentesis.

The certainty of evidence was downgraded mainly due to the risk of bias in the included studies and imprecision of the pooled estimates due to the limited number and size of RCTs included. Although it was not possible to carry out subgroup analyses according to the type of internal TMJ disorder to test this assumption as a potential source of heterogeneity or effect modifiers, pairwise comparisons did not show heterogeneity and, therefore, did not downgrade the certainty of evidence.

The effect of corticosteroids on pain is due to their anti-inflammatory effect through the inhibition of the production and secretion of proinflammatory cytokines, such as interleukins, tumor necrosis factor-alpha, interferon-gamma, and factor-stimulating granulocytic and macrophage colonies through direct interference in cascades and genomic mechanisms. They also inhibit the accumulation of macrophages and neutrophils in inflammatory foci because they repress the expression of endothelial adhesion molecules and the synthesis of the plasminogen activator [58,59].

Although our results agree with evidence based on other joints regarding the effectiveness of corticosteroids in pain management [60,61], previous NMAs suggested that intra-articular injections of corticosteroids had no effect on improving temporomandibular joint pain [12,62]. The latter grouped all types of corticosteroids in the same node, whether combined with arthrocentesis or not, presuming that all corticosteroids have the same action mechanism and that arthrocentesis has no effect on symptomatology. Because arthrocentesis has an effect on the joint by removing inflammatory cells, freeing the articular disc from adhesions and fibrillations, and allowing adequate translation of the condyle [63,64], assuming that arthrocentesis has no effect could mask the real effect of corticosteroids.

Our results showed no convincing evidence that corticosteroids improve function compared to arthrocentesis, which is consistent with other systematic reviews for this and other bodies’ articulations [60,65]. It is important to keep in mind that arthrocentesis is not innocuous and could reduce the pain and increase the maximum mouth opening [64], which could explain why corticosteroids were not shown to be better than arthrocentesis in improving the RoM.

Because the different corticosteroids have various mechanisms of action and times of action, it is essential for the clinician to know which presents better clinical results in the short (1 to 3 months), medium (4 to 6 months), and long term (>6 months). Corticosteroids suitable for intra-articular administration are the particulates, which are water-insoluble and form microcrystalline particles, so they have a slower and longer-lasting release (triamcinolone, methylprednisolone, or hydrocortisone), and non-particulates that are water-soluble, which have a rapid but shorter-lasting effect (dexamethasone and betamethasone sodium phosphate) [58]. Unfortunately, the available evidence is particularly lacking for long-term outcome data to know the performance of corticosteroids in this time frame. Although most of the studies included in this review evaluated the short- and medium-term effectiveness, most comparisons are based on indirect evidence or a single study, hindering decision making. The need for long-term data (1 or 2 years) has been highlighted in other systematic reviews [60], limiting decision making in the chronic course of TMJ disorder.

Several included studies evaluated whether the effectiveness of corticosteroids is enhanced when combined with arthrocentesis or other therapies, such as intra-articular injection of hyaluronic acid. Our results showed that the effect of corticosteroids tends to be diluted when combined with other substances. Nevertheless, the low certainty of evidence should be considered when interpreting the results.

A recent systematic review showed that TMJ disorder significantly impacts oral-health-related quality of life [66]. However, QoL was poorly addressed in the included studies to reach any meaningful conclusions. Because pain and functional limitations negatively impact the overall the QoL of patients with TMDs, studies should prioritize this outcome when assessing the effectiveness of corticosteroid injections.

### Strengths and Limitations of the Review Process

This is the first systematic review that explores the effectiveness of corticosteroids alone or combined with other substances for any internal TMJ disorders. Considering that various corticosteroids have different mechanisms of action and duration, grouping them all into a single node makes it impossible to know which one has the best effect in the short, medium, and long term.

Another strength of our review was that we evaluated the effectiveness of corticosteroids both separately and combined with other minimally invasive therapies. Some previous systematic reviews did not distinguish if the corticosteroids were injected alone or combined with other therapies, such as arthrocentesis, which can lead to underestimation or overestimation of the effect of the corticosteroids [67,68].

We rigorously followed Cochrane’s methodology and PRISMA guidelines to render this systematic review reliable and reproducible.

Finally, our review included the GRADE framework for drawing conclusions from the NMA that facilitate decision making by classifying interventions in groups from the most to the least effective, considering the magnitude of effect and the certainty of the evidence.

This review process is not exempt from limitations. We included RCTs comparing corticosteroids with other minimally invasive interventions to obtain all indirect evidence to allow us to increase the number of comparisons between corticosteroids. Consequently, many estimates came only from indirect evidence, which can explain the low or very low certainty of evidence across most comparisons.

## 5. Conclusions and Implications for Practice and Research

Betamethasone and arthrocentesis plus dexamethasone are likely the most effective in managing short- and medium-term pain when compared with arthrocentesis. However, we can draw no conclusion about the range of motion or QoL outcomes because of the low and very low certainty of evidence. This makes decision making concerning managing symptomatic internal TMJ disorders with corticosteroids strongly dependent on the clinician’s clinical experience and patient preferences, considering various factors, such as costs, feasibility, and acceptability.

Future RCTs should be designed with large sample sizes, longer follow-up periods, and measuring QoL as an outcome of interest. In addition, RTCs should compare the different types of corticosteroids alone (without combinate) and with arthrocentesis as the comparator to establish the additional benefits of corticosteroids over the standard reference.

## 6. Differences between Protocol and Review

We planned to measure all outcomes at 1, 3, 6, and 12 months. However, due to inconsistency in the time points reported by trials, we used the following ranges: 1 month; 3 months: studies that reported outcomes between 2 and 3 months of follow-up; 6 months: studies that reported outcomes between 4 and 6 months of follow-up; and 12 months: studies that reported outcomes between 7 and 12 months of follow-up.

We had planned to conduct subgroup analyses based on the duration of follow-up (short duration (<6 months) vs. long duration (>six months)). Instead, we presented the outcomes of interest at different time points.

We planned to complete the review on 31 July 2020; because the review took more time, we updated the search on 31 December 2023 to keep it up to date.

## Figures and Tables

**Figure 1 jcm-13-04557-f001:**
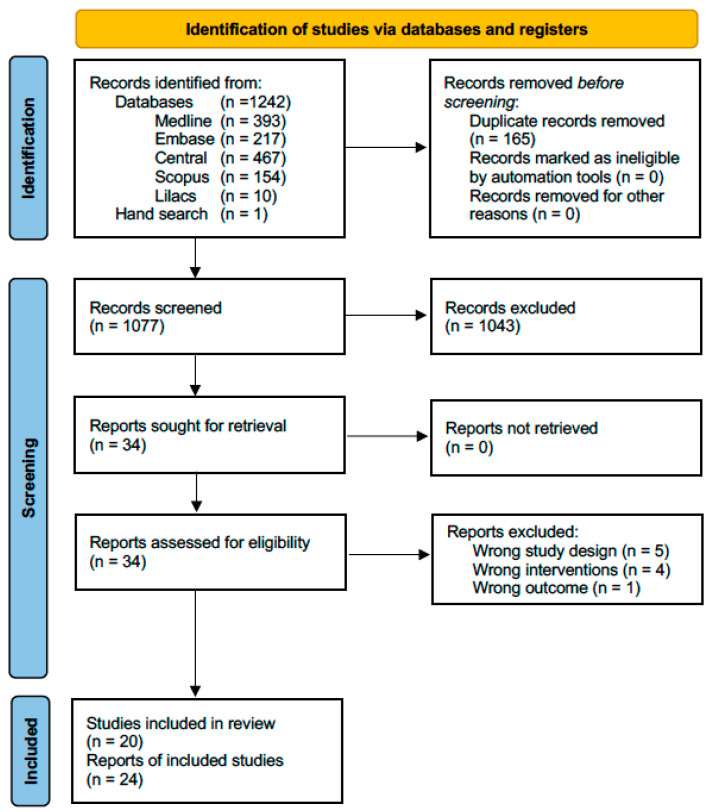
Flow chart of systematic literature review.

**Figure 2 jcm-13-04557-f002:**
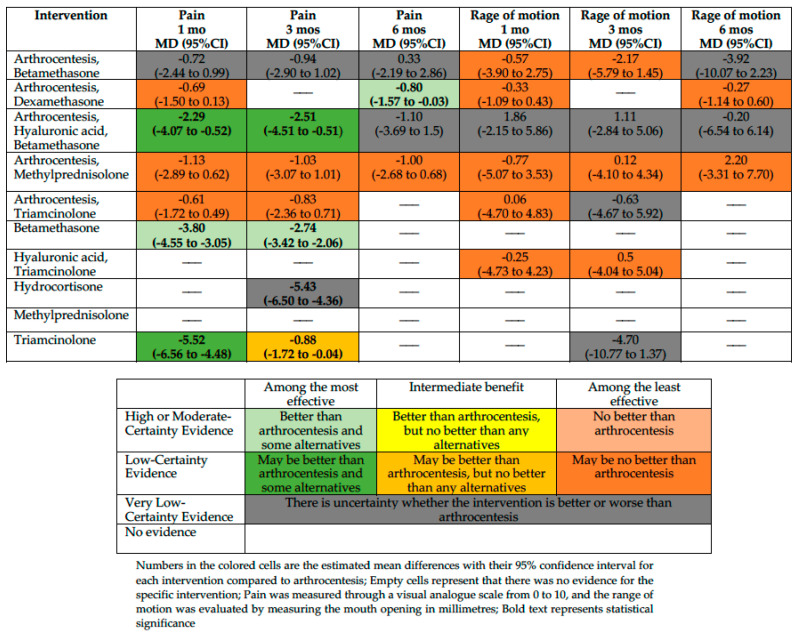
Summary of effects compared with arthrocentesis.

**Figure 3 jcm-13-04557-f003:**
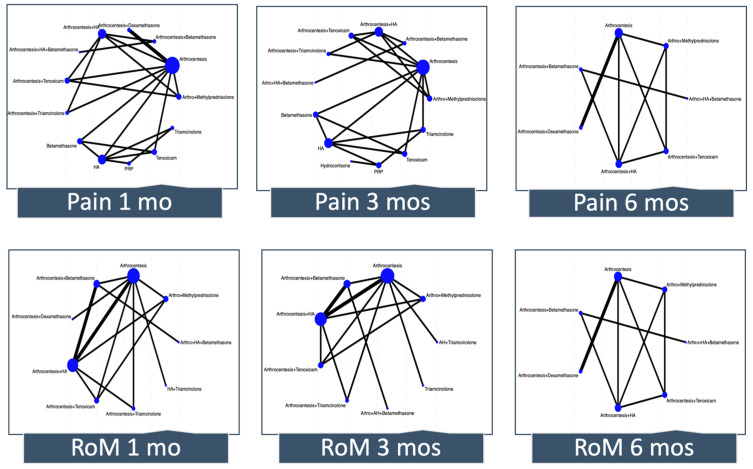
Network meta-analysis plots. Each node (circle) indicates the type of corticosteroid therapy, and its size is proportional to the number of patients who received the treatment. Each solid line connecting two nodes indicates a direct comparison between two interventions, and the thickness of each solid line is proportional to the number of trials directly comparing the two modalities. RoM, range of motion; HA, hyaluronic acid; PRP; platelet-rich plasma.

**Table 1 jcm-13-04557-t001:** Characteristics of included studies.

Study [Ref]	Study Design	Country	Setting	N	TMJID	Sex M/F	Age Mean (SD)	Follow-Up	Intervention	Outcomes Reported
AbdukRazzal 2020 [34]	Parallel group	Iraq	Hospital	30	Internal derangement	4/26	Range 18 to 42 years	1 and 4 months	- Intervention 1: 1 mL of methylprednisolone - Intervention 2: arthrocentesis	Pain Range of motion
Bjørnland 2007 [35,36,37]	Parallel group	Norway	University	40	Osteoarthritis	6/34	51.7 (13.05)	1 and 6 months	- Intervention 1: 0.7–1 mL of betamethasone - Intervention 2: 0.7–1 mL of hyaluronic acid	Pain Range of motion Adverse effect
Bouloux 2016 [38,39]	Parallel group	USA	University	102	Disc displacement with or without reduction and osteoarthritis	13/89	42.0 (17.83)	3 months	- Intervention 1: 200 mL arthrocentesis plus 1 mL of betamethasone - Intervention 2: 200 mL arthrocentesis plus 1 mL of hyaluronic acid - Intervention 3: 1 mL of lactated Ringer’s	Pain Range of motion Quality of life Adverse effect
Comert-Kiliç 2016 [40]	Parallel group	Turkey	University	24	Osteoarthritis	3/21	33.8 (12.3)	12 months	- Intervention 1: 1 mL of methylprednisolone plus 100 mL arthrocentesis - Intervention 2: arthrocentesis	Pain Range of motion Adverse effect
Dharamsi 2022 [41]	Parallel group	India	University	40	Internal derangement	17/23	37.0 (NR)	1 and 3 months	- Intervention 1: 100 mL arthrocentesis plus 20 mg of hyaluronic acid - Intervention 2: 100 mL arthrocentesis plus 20 mg of triamcinolone	Pain Range of motion
Dolwick 2020 [42,43]	Parallel group	USA	University	24	Disc displacement with and without reduction, osteoarthritis	0/24	Range 18 to 80 years	2 weeks, 6 weeks, and 3 months	- Intervention 1: 100 mL arthrocentesis plus 20 mg of triamcinolone - Intervention 2: 100 mL arthrocentesis	Pain Range of motion
Gencer 2014 [44]	Parallel group	Turkey	University	100	Osteoarthritis	45/55	42.5 (10.2)	1 and 5 months	- Intervention 1: 7 mg of betamethasone - Intervention 2: 10 mg of hyaluronic acid - Intervention 3: 20 mg of tenoxicam - Intervention 4: saline solution	Pain
Giraddi 2012 [45]	Parallel group	India	University	16	Internal derangement	6/10	26.4 (5.22)	1, 2, and 6 months	- Intervention 1: 100–200 mL arthrocentesis plus 1 mL of betamethasone - Intervention 2: 100–200 mL arthrocentesis plus 1 mL of hyaluronic acid	Pain Range of motion
Giraddi 2015 [46]	Parallel group	India	University	14	Disc displacement with or without reduction	8/6	30.4 (6.94)	1, 2, and 6 months	- Intervention 1: 100–200 mL arthrocentesis plus 1 mL of betamethasone - Intervention 2: 100–200 mL arthrocentesis plus a combination of 0.5 mL of betamethasone and 0, 5 mL of hyaluronic acid	Pain Range of motion
Gokce-Kutuk 2019 [47]	Parallel group	Turkey	Hospital	31	Osteoarthritis	10/21	36.4 (8.7)	1, 3, and 6 months	- Intervention 1: 1 mL of triamcinolone, 1 mL of platelet-rich plasma - Intervention 2: 1 mL of hyaluronic acid	Pain
Gupta 2018 [48]	Parallel group	India	University	20	Disc displacement with reduction	NR	NR	1 week and 3 months	- Intervention 1: 0, 6 mL of platelet-rich plasma - Intervention 2: 0, 5 mL of hydrocortisone	Pain Range of motion
Huddleston-Slater 2012 [49]	Parallel group	Netherlands	University	28	Arthralgia	5/23	33.9 (15.03)	3 weeks and 6 months	- Intervention 1: 300 mL arthrocentesis plus 1 mL of dexamethasone - Intervention 2: arthrocentesis	Pain Range of motion Quality of life
Isacsson 2019 [50]	Parallel group	Sweden	Hospital	54	Arthralgia	10/44	52 (17.9)	1 month	- Intervention 1: 1 mL of methylprednisolone - Intervention 2: arthrocentesis	Pain Range of motion Quality of life Adverse effect
Kopp 1985 [51]	Parallel group	Sweden	University	33	Osteoarthritis	4/29	46.4 (13.4)	1 month	- Intervention 1: 0.5 mL of betamethasone - Intervention 2: 0.5 mL of hyaluronic acid	Pain Range of motion
Manfredini 2012 [52]	Parallel group	Italy	University	60	Osteoarthritis	9/51	50.1 (NR)	3 and 6 months	- Intervention 1: 300 mL arthrocentesis, arthrocentesis plus 1 mL of triamcinolone - Intervention 2: arthrocentesis 300 mL plus 1 mL of low-molecular-weight hyaluronic acid	Pain Range of motion
Majeed 2020 [53]	Parallel group	Iraq	Hospital	54	Disc displacement with reduction	9/45	Range 18.2 to 55.0	1, 3, 6, and 12 months	- Intervention 1: arthrocentesis plus 1 mL of betamethasone - Intervention 2: arthrocentesis plus 1 mL of hyaluronic acid	Pain Range of motion
Marzook 2020 [54]	Parallel group	Egypt	University	16	Disc displacement with reduction	NR	NR	1 and 3 months	- Intervention 1: 0.5 mL of triamcinolone plus 0.5 mL of hyaluronic acid - Intervention 2: arthrocentesis	Pain Range of motion Adverse effect
Singh 2022 [55]	Parallel group	India	University	20	Internal derangement	NR	NR	3 months	- Intervention 1: 10 mg of triamcinolone - Intervention 2: 100 mL arthrocentesis	Pain Range of motion
Tabrizi 2014 [56]	Parallel group	Iran	University	60	Internal derangement	13/47	27.5 (7.25)	1 and 6 months	- Intervention 1: 200 mL arthrocentesis plus 8 mg of dexamethasone - Intervention 2: 200 mL arthrocentesis	Pain Range of motion
Yapici-Yavuz 2018 [57]	Parallel group	Turkey	University	44	Disc displacement without reduction	6/38	NR	1, 3, and 6 months	- Intervention 1: 200 mL arthrocentesis plus methylprednisolone - Intervention 2: 200 mL arthrocentesis plus hyaluronic acid - Intervention 3: 200 mL arthrocentesis plus tenoxicam - Intervention 4: 200 mL arthrocentesis	Pain Range of motion Adverse effect

NR: Not reported; TMJID: Temporomandibular joint internal disorders.

## Data Availability

The data and materials supporting the conclusions of this manuscript are included in the article.

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
