# Peer review of "Corticosteroids for the Treatment of Internal Temporomandibular Joint Disorders: A Systematic Review and Network Meta-Analysis"

_jcm, 2024, doi:10.3390/jcm13154557_

Round 1

Reviewer 1 Report

Comments and Suggestions for Authors

Very well structured Systematic review with a topic that brings interest in all Dentistry.

Few considerations:

Ensure consistency in reporting (e.g., using “NMA” consistently instead of alternating with “network meta-analysis”).

Consider providing a flow diagram (PRISMA flowchart) summarizing the study selection process, which can help readers quickly understand the number of studies included at each stage.

Overall, an exciting paper about TMJ.

Author Response

General comment: Very well-structured Systematic review with a topic that brings interest in all Dentistry. Overall, an exciting paper about TMJ.

Response: We very much appreciate the positive comments by reviewer #1 and will introduce the suggested modifications (kindly see responses to specific comments). Additions to the manuscript in response to comments are highlighted in blue to facilitate your review.

Specific comment #1: Ensure consistency in reporting (e.g., using “NMA” consistently instead of alternating with “network meta-analysis”).

Response: Thank you for the remarks. We have used NMA consistently throughout the manuscript

Specific comment #2: Consider providing a flow diagram (PRISMA flowchart) summarizing the study selection process, which can help readers quickly understand the number of studies included at each stage.

Response: Figure 1 corresponds to a PRISMA flowchart and you can find in the main manuscript.

Reviewer 2 Report

Comments and Suggestions for Authors

Dear Authors,

Thank you for taking the time and effort to conduct this fascinating research on corticosteroids in treating internal temporomandibular joint (TMJ) disorders. This well-written and well-organized study (a systematic review and network meta-analysis) comprehensively evaluated corticosteroid injections for treating internal TMJ disorders. I believe this study satisfies all the quality norms and will add significant value to the existing body of literature.

Best Wishes!

Reviewer

Comments on the Quality of English Language

Punctuation check is needed.

Author Response

General comment: Thank you for taking the time and effort to conduct this fascinating research on corticosteroids in treating internal temporomandibular joint (TMJ) disorders. This well-written and well-organized study (a systematic review and network meta-analysis) comprehensively evaluated corticosteroid injections for treating internal TMJ disorders. I believe this study satisfies all the quality norms and will add significant value to the existing body of literature.

Response: We very much appreciate the positive comments by reviewer #2.

Reviewer 3 Report

Comments and Suggestions for Authors

I am grateful for the opportunity to review the article Corticosteroids for the Treatment of Internal Temporomandibular Joint Disorders: A Systematic Review and Network Meta-Analysis submitted to the JCM MDPI journal.

The article has a correct writing structureand and methodology, and it is generally well prepared.

I would recommend that the authors consider two points. Just a cosmetic editorial change: On page five, when the number of studies is mentioned, only one is mentioned in numerical form, 8 of osteoarthritis, and I think it should be in full. 

The article's discussion, while not a hindrance to publication, has the potential to significantly enrich the field. I believe that a more comprehensive discussion of the pharmacology of the products could be beneficial, going beyond a simple review of the results.

Author Response

General comment: I am grateful for the opportunity to review the article Corticosteroids for the Treatment of Internal Temporomandibular Joint Disorders: A Systematic Review and Network Meta-Analysis submitted to the JCM MDPI journal.

The article has a correct writing structure and methodology, and it is generally well prepared.

Response: We very much appreciate the positive comments by reviewer #3 and will introduce the suggested modifications (kindly see responses to specific comments). Additions to the manuscript in response to comments are highlighted in blue to facilitate your review.

Specific comment #1: I would recommend that the authors consider two points. Just a cosmetic editorial change: On page five, when the number of studies is mentioned, only one is mentioned in numerical form, 8 of osteoarthritis, and I think it should be in full.

Response: We thank the reviewer for noticing this typo, and we corrected it.

Specific comment #2: The article's discussion, while not a hindrance to publication, has the potential to significantly enrich the field. I believe that a more comprehensive discussion of the products' pharmacology could be beneficial, going beyond a simple review of the results.

Response: We thank the reviewer for this suggestion. To better understand the results, we added the mechanisms of action of corticosteroids to the discussion.

Discussion, lines 336-342: “The effect the corticosteroid on pain is due to its anti-inflammatory effect through the inhibition of the production and secretion of proinflammatory cytokines such as interleukins, tumour necrosis factor-alpha, interferon-gamma and factor-stimulating granulocytic and macrophage colonies by direct interference on cascades and genomic mechanisms. They also inhibit the accumulation of macrophages and neutrophils in inflammatory foci because they repress the expression of endothelial adhesion molecules and the synthesis of the plasminogen activator”

Discussion, lines 359-366:Since the different corticosteroids have various mechanisms of action and time of action, it is essential for the clinician to know which presents better clinical results in the short (1 to 3 months), medium (4 to 6 months), and long-term (>6 months). Corticosteroids suitable for intra-articular administration are the particulates which are water-insoluble and form microcrystalline particles, so they have a slower and longer-lasting release (triamcinolone, methylprednisolone, or hydrocortisone) and non-particulates, water-soluble, which have a rapid but shorter-lasting effect (dexamethasone, betamethasone sodium phosphate). Unfortunately, the available evidence is particularly lacking for long-term outcomes data to know the performance of corticosteroids in this time frame…”

Reviewer 4 Report

Comments and Suggestions for Authors

  1. I kindly thank you for the opportunity to review. I have the following comments on the work:

  2. Line 1 – change 'Review' to 'Systematic Review'.
  3. Please add the country to the authors' first affiliation.
  4. Abstract – unfortunately, the abstract is not written according to the PRISMA abstract guidelines. Please revise its entire structure and attach the guidelines as additional material. https://www.prisma-statement.org/abstracts
  5. TMD – throughout the work, please change the abbreviation to TMDs – this emphasizes it as 'disorders' rather than a single disorder. https://www.ncbi.nlm.nih.gov/books/NBK557995/
  6. Lines 44-45 – Please add more information on the prevalence of TMDs. Please add that TMDs are estimated at 34%. Please refer to https://doi.org/10.3390/jcm13051365. Additionally, please describe the differences in prevalence depending on the continent and highlight the gender differences – the ratio of women to men. In reference to the new meta-analysis.
  7. Line 46 – 'counseling' – this description is too vague. Please clarify.
  8. Line 48 – please add information about treatment with botulinum toxin as well - https://www.ncbi.nlm.nih.gov/pmc/articles/PMC5634354/
  9. Line 54 – 'joints[7]. ' – add a space.
  10. Lines 56-58 – this is not supported by a citation, so it is an opinion rather than a fact. Please add citations.
  11. Line 70 – instead of citation number 13, it should be 10.1136/bmj.n7 according to PRISMA guidelines.
  12. Line 72 – 'CRD4201912914' – I cannot verify this number in the PROSPERO database. A message appears – 'No hits for line : CRD4201912914'. I checked publication number 14. There is an error in the number, it should be 'CRD42019129014'. Please ensure accuracy in writing such important numbers for the work.
  13. Can the authors explain why the following authors are listed in the registered protocol 'Daniela Torres, Carlos Zaror, Verónica Iturriaga', while the protocol publication has the following authors 'Daniela Torres, Carlos Zaror, Verónica Iturriaga, Aurelio Tobias', and in the submitted work to JCM, even more authors were added? Why such discrepancies in authorship?
  14. Line 83 – please add information regarding the publication time range that the authors included. Did the authors search the databases from their inception?
  15. Lines 84-85 – 'We conducted electronic searches in MEDLINE, CENTRAL, EMBASE, SCOPUS, and LILACS through December 31, 2023' – however, the PROSPERO registration lists the following dates 'Anticipated completion date [1 change] 31 July 2020'. Why these differences? Please explain.
  16. Line 101 – please add the access date to the link.
  17. Line 104 – please replace initials with 'Two reviewers'.
  18. 2.7. Data synthesis – how was the homogeneity or heterogeneity of the studies checked?
  19. Line 134 – where can I find these charts?
  20. Line 142 – 'high risk of bias' – please describe more precisely how the authors did this. On what basis?
  21. Line 146 – 'low[18, 19].' – add a space.
  22. Figure 1. – the chart is quite unclear in terms of numbers. The first box is the most unclear, the authors write 'Records identified from*:' – what does the asterisk refer to? Furthermore, the last box shows two values, n-20 and n-24. I understand that 20 studies include 24 descriptions of studies, right? But in fact, 20 works were included and this should be captured, with an explanation in the text.
  23. Please add a list of studies excluded at the 'Reports excluded' stage. In the main text according to PRISMA recommendations.
  24. Table 1. – please cite the authors chronologically or alphabetically – the first column.
  25. Table 1. – why for the author 'Bjørnland 2007 [23, 39, 40]' and 'Bouloux 2016 [28, 41]' are there several citations? Considering that, for example, works 39 and 40 do not refer to Bjørnland.
  26. Table 1. – explain all abbreviations from the table under the table.
  27. Line 196 – 'management[29, 30],' – please add spaces. Continue to correct this throughout the text. There should be a space between the citation and the end of the sentence.
  28.  
    1. Discussion – I would ask to add more anatomical and physiological references that could explain the obtained results.
  29. Additionally, please provide information on whether the results were homogeneous or heterogeneous.
  30. Please correct the bibliography according to the journal's requirements. Bold the year, add missing information about the works.
  31. Please ensure that the works cited in the introduction are current literature, not from at least 15 years ago, for example, works 1, 2, 3.
  32. Additional materials:
  33. Supplement 3: – the number 'N' for the Wenneberg 1978 study is missing.
  34. Supplement 6: Network meta-analysis plots – I would suggest adding them to the main text as they significantly enrich the work.
  35. Supplement 4: – add bibliography under the material.
  36. Supplement 5: – add bibliography under the material.

Reviewer 5 Report

Comments and Suggestions for Authors

Thank you for the opportunity to review this article.

few comments:

  • Some sections lack detailed explanation of the underlying mechanisms, which might be necessary for a thorough understanding of the treatment's efficacy and potential side effects.
  • There is a need for more critical analysis of the limitations of the included studies, such as potential biases and variations in study design.
  • Some methodological details, such as the handling of missing data and the choice of statistical models, could be elaborated further
  • The results section could benefit from a more detailed breakdown of the findings, perhaps with additional tables or figures to illustrate key points.
  • There are a few instances of jargon that might be confusing to readers unfamiliar with the field. Providing brief definitions or explanations could improve readability.
  • Some references are missing details such as page numbers or DOI links. Ensuring complete citation information would enhance the paper's scholarly rigor.

Author Response

General comment: Thank you for the opportunity to review this article.

Response: We very much appreciate the positive comments by reviewer #5 and will introduce the suggested modifications (kindly see responses to specific comments). Additions to the manuscript in response to comments are highlighted in blue to facilitate your review.

Specific comment #1: Some sections lack detailed explanation of the underlying mechanisms, which might be necessary for a thorough understanding of the treatment's efficacy and potential side effects.

Response: We thank the reviewer for this comment. To better understand the results, we added the mechanisms of action of corticosteroids to the discussion.

Discussion, lines 337-343: “The effect the corticosteroid on pain is due to its anti-inflammatory effect through the inhibition of the production and secretion of proinflammatory cytokines such as interleukins, tumour necrosis factor-alpha, interferon-gamma and factor-stimulating granulocytic and macrophage colonies by direct interference on cascades and genomic mechanisms. They also inhibit the accumulation of macrophages and neutrophils in inflammatory foci because they repress the expression of endothelial adhesion molecules and the synthesis of the plasminogen activator”

Discussion, lines 360-367:Since the different corticosteroids have various mechanisms of action and time of action, it is essential for the clinician to know which presents better clinical results in the short (1 to 3 months), medium (4 to 6 months), and long-term (>6 months). Corticosteroids suitable for intra-articular administration are the particulates which are water-insoluble and form microcrystalline particles, so they have a slower and longer-lasting release (triamcinolone, methylprednisolone, or hydrocortisone) and non-particulates, water-soluble, which have a rapid but shorter-lasting effect (dexamethasone, betamethasone sodium phosphate). Unfortunately, the available evidence is particularly lacking for long-term outcomes data to know the performance of corticosteroids in this time frame…”

Specific comment #2: There is a need for more critical analysis of the limitations of the included studies, such as potential biases and variations in study design.

Response: Following the reviewer’s suggestion, we discussed how the included studies' limitations affected the certainty of evidence.

Discussion, lines 331-336: “The certainty of evidence was downgraded mainly due to the risk of bias in the included studies and imprecision of the pooled estimates due to the limited number and size of RCTs included. Although it was not possible to carry out subgroup analyses according to the type of internal TMJ disorder to test this assumption as a potential source of heterogeneity or effect modifiers, pairwise comparisons did not show heterogeneity and, therefore, did not downgrade the certainty of evidence.

Specific comment #3: Some methodological details, such as the handling of missing data and the choice of statistical models, could be elaborated further

Response: Following the reviewer’s suggestion, we have added details about missing data and model.

Material and methods, Data synthesis, lines 126-128: We conducted frequentist fixed-effects NMA using multivariate meta-analysis using Stata 18 (Stata Corp LP, USA) because we assumed that all studies share a single common effect since we had few studies for each comparison”

Material and methods, Data synthesis, lines 148-149: “The original authors of the study were contacted to obtain the missing data and details of any outcomes that may have been measured but not reported. We did not use any other statistical methods or perform any further imputation to account for missing data”

Material and methods, Data synthesis, lines 151-156: We assessed the clinical heterogeneity in the included studies by examining the similarity between the types of participants, interventions and outcomes. In pairwise meta-analyses, we estimated the statistical heterogeneity for each comparison using the I² statistic, where values over 50% indicate considerable heterogeneity. In the network meta-analysis, we assumed a common estimate for the heterogeneity variance across all comparisons”

Specific comment #4: The results section could benefit from a more detailed breakdown of the findings, perhaps with additional tables or figures to illustrate key points.

Response: We thank the reviewer for this comment. We have strongly followed the PRISMA reporting guidelines for network meta-analysis and reported everything that is necessary according to these guidelines. Due to the limited word count of the scientific manuscripts, we have prioritized highlighting only the most important aspects for users in the main text and reported other details, including additional tables and figures, in the appendices.

Specific comment #5: There are a few instances of jargon that might be confusing to readers unfamiliar with the field. Providing brief definitions or explanations could improve readability.

Response: Following the reviewer’s comment, we have added definitions of some terms, such as counseling, and arthrocentesis. We will be happy to add more explanations if the reviewer clarifies to which specific terms they are referring.

Specific comment #6: Some references are missing details such as page numbers or DOI links. Ensuring complete citation information would enhance the paper's scholarly rigor.

Response: Following the reviewer’s comment, we have corrected the bibliography according to the journal's requirements.

Round 2

Reviewer 4 Report

Comments and Suggestions for Authors

Thank you for resubmitting the work. I am impressed with the revisions and responses made by the authors. I accept the work in its current form.

Best regards